# Effect of a Rumen-Protected Microencapsulated Supplement from Linseed Oil on the Growth Performance, Meat Quality, and Fatty Acid Composition in Korean Native Steers

**DOI:** 10.3390/ani11051253

**Published:** 2021-04-27

**Authors:** Chae-Hyung Sun, Jae-Sung Lee, Jalil Ghassemi Nejad, Won-Seob Kim, Hong-Gu Lee

**Affiliations:** Department of Animal Science and Technology, Konkuk University, Seoul 05029, Korea; cocosch1114@gmail.com (C.-H.S.); foodleeking@gmail.com (J.-S.L.); jalilgh@konkuk.ac.kr (J.G.N.); kws9285@hanmail.net (W.-S.K.)

**Keywords:** rumen-protected microencapsulated linseed oil, omega-3 fatty acids, Korean native steers, growth performance, meat quality

## Abstract

**Simple Summary:**

In vitro and in vivo studies on the supplementation of rumen-protected microencapsulated fatty acid from linseed oil (MO) on rumen digestibility, physiological profile, growth performance, meat quality, and meat fatty acid profile in Korean native steers were conducted. The in vitro study showed that 3% MO is an optimal dose, as there were decreases in the neutral detergent fiber and acid detergent fiber digestibility at 48 h. Supplementation with 3% MO not only promotes growth performance but also enhances the omega-3 fatty acid concentration of meat in Korean native steers.

**Abstract:**

We evaluated the effects of a rumen-protected microencapsulated supplement from linseed oil (MO) on ruminal fluid, growth performance, meat quality, and fatty acid composition in Korean native steers. In an in vitro experiment, ruminal fluid was taken from two fistulated Holstein dairy cows. Different levels of MO (0%, 1%, 2%, 3%, and 4%) were added to the diet. In an in vivo experiment, eight steers (average body weight = 597.1 ± 50.26 kg; average age = 23.8 ± 0.12 months) were assigned to two dietary groups, no MO (control) and MO (3% MO supplementation on a DM basis), for 186 days. The in vitro study revealed that 3% MO is an optimal dose, as there were decreases in the neutral detergent fiber and acid detergent fiber digestibility at 48 h (*p* < 0.05). The in vivo study showed increases in the feed efficiency and average daily gain in the 3% MO group compared to the control group on days 1 to 90 (*p* < 0.05). Regarding meat quality, the shear force produced by the longissimus thoracis muscle in steers from the 3% MO group was lower than that produced by the control group (*p* < 0.05). Interestingly, in terms of the fatty acid profile, higher concentrations of C22:6n3 were demonstrated in the subcutaneous fat and higher concentrations of C18:3n3, C20:3n3, and C20:5n3 were found in the intramuscular fat from steers fed with 3% MO (*p* < 0.05). Our results indicate that supplementation with 3% MO supplements improves the growth performance and meat quality modulated by the omega-3 fatty acid content of meat in Korean native steers.

## 1. Introduction

The purpose of improving meat quality is to promote the nutritional and commercial value of beef. The intramuscular fat content of beef is an important factor in quality evaluation, as it greatly affects the taste, juiciness, flavor, and tenderness of beef [1]. In addition, to increase the content of intramuscular fat in beef, the use of grain-based feed is encouraged. However, recently, negative perceptions about fat in beef have gradually been spreading. It has been reported that the use of grain-based feed for ruminants induces increases in the content of omega-6 fatty acids in beef, causing an imbalance in the omega-6/omega-3 ratio [2]. It has been previously documented that the consumption of foods with an unbalanced omega-6/omega-3 ratio is associated with various diseases such as obesity, diabetes, cancer, and high blood pressure [3,4]. Accordingly, research on enrichment with omega-3 to improve the balance of omega-6/omega-3 in beef is actively being conducted. Therefore, to maintain an adequate amount of intramuscular fat, it is necessary to increase the content of omega-3 fatty acids, which are beneficial for human health, and to reduce the omega-6/omega-3 ratio.

The strategies for increasing the content of omega-3 in beef include the use of forage feeding [5,6], rumen-unprotected supplementation [7,8], rumen-protected supplementation [9,10], and oil seed feeding [11]. Among these, supplementation with linseed oil (enriched omega-3 fatty acid), which is not protected in the rumen, has been shown to increase the omega-3 fatty acid concentration in plasma and fat tissues [12]. Meanwhile, a loss of omega-3 fatty acids was found to occur through extensive biohydrogenation within the rumen. Moreover, soybean oil was shown to adversely affect rumen fermentation, causing negative effects, such as a reduction in productivity [13]. To prevent these problems, rumen-protected supplements have been developed in various ways.

Although many studies have investigated rumen-protected supplementation [9,10], the effects on performance, meat quality, and fatty acid composition in Korean native steers require further research. Therefore, the objective of this study was to conduct an in vitro experiment to determine the optimal dosage of rumen bypass microencapsulated fatty acids from linseed oil that does not negatively affect ruminal fermentation. Based on the results of the in vitro experiment, we investigated the effect of supplementation with rumen bypass microencapsulated fatty acids from linseed oil on the growth performance, physiological indicators, meat quality, and fatty acid composition in Korean native steers.

## 2. Materials and Methods

All experimental procedures involving animals were performed according to the Animal Experimental Guidelines provided by the Animal Care and Use Committee of Konkuk University, Republic of Korea (Approval number, KU19146).

### 2.1. In Vitro Batch Culture

#### 2.1.1. Experimental Materials and Methods

Ruminal fluid was taken from two fistulated Holstein dairy cows 2 h before feeding. The cows were fed tall fescue and concentrate (a ratio of 6:4) once a day at 0.900 h. Water was supplied ad libitum. Ruminal fluid was filtered with two layers of cheese cloth and mixed in the same ratio. The basal diet fed to the cows was milled through a 1 mm screen and used as a substrate in a ratio of 6:4. Samples of the basal diet (0.5 g) were put in ANKOM bags (filter bag 58, ANKOM Tech, Macedon, NY, USA) to analyze the digestibility. Menke’s buffer solution [14] adjusted to pH 6.8 was prepared under continuous flushing with CO_2_ at 39 °C. Then, buffer solution and ruminal fluid were mixed in a ratio of 3:1. A quantity of 250 mL of buffered ruminal fluid was filled into the Erlenmeyer flask and added to five ANKOM bags. CO_2_ gas was flushed into the headspace of the flask. The experiment was conducted on three replicates of five treatments, and samples were incubated in a shaking incubator (JSSI-300C, JSP Corp, Gongju, Korea) for 48 h at 39 °C.

The preparation of rumen bypass microencapsulated fatty acids from linseed oil (MO; Microtinic^®^ Omega) was obtained from Vetagro S.p.A (Reggio Emilia, Italy). The material was prepared via a patented spray-cooling microencapsulation process. Briefly, after heating the hydrogenated palm oil, linseed oil was mixed with hydrogenated palm oil and an active principle. A mixture of materials was moved to a chilling chamber and then cut to a size of 1000 to 1500 microns. Finally, the MO was sealed and stored at 4 °C. The ingredients of the rumen-protected microencapsulated fatty acid from linseed oil (MO) were linseed oil (35%), vitamin E (0.5%), rosemary extract (0.3%), and hydrogenated palm oil (64.2%). The main components of the MO were crude fat (95%) and ash (5%) [15].

#### 2.1.2. Analysis

The pH, ammonia-N, volatile fatty acid (VFA) content, and digestibility of the ruminal fluid were analyzed. Measurement of the ruminal pH was conducted with a pH meter (S20 SevenEasy pH, Mettler Toledo, Greifensee, Switzerland), and 5 mL residual ruminal fluid samples were mixed with 1 mL of 2% HgCl_2_ (wt/vol) and 25% meta-phosphoric acid (wt/vol) solution for ammonia-N and VFA analysis. Four milliliters of culture solution was collected for the ammonia (1 mL) and VFA (3 mL) analyses. All samples were kept frozen at –20 °C until analysis. Ammonia-N was determined via the method presented by Chaney and Marbach [16] using spectrophotometry (Synergy2; Biotek Instruments, Winooski, VT, USA). To determine the VFA content, 3 mL of the culture solution was centrifuged at 20,000× *g* for 20 min at 4 °C, and 1 mL of supernatant was collected. The 1 mL supernatant sample was mixed with 50 μL of 2% pivalic acid (wt/vol) and used as an internal standard. The VFA composition was analyzed using a gas chromatograph (HP 6890, Agilent Technologies, Santa Clara, CA, USA) equipped with a flame ionization detector and a capillary column (DB-FFAP; Agilent Technologies). The chemical analyses were conducted using the standard method for the association of official analytical chemists (AOAC) [17]. The dry matter (DM) content was determined using the ANKOM bag, and the ANKOM bag was weighed after drying for 48 h at 60 °C in a dry oven. The digestibility of DM was calculated as follows:DM digestibility (%) = dried residue sample/sample weight × 100.(1)

The neutral detergent fiber (NDF) and acid detergent fiber (ADF) contents were determined using an ANKOM bag technique. Feed samples (0.5 g) were put into ANKOM bags. The bags were immersed in acetone for 10 min, dried for 10 min, and then put into an ANKOM 200 fiber analyzer (ANKOM Tech, Macedon). The NDF or ADF solution, 4 mL of α-amylase (not included in the ADF analysis), and 20 g of anhydrous sodium sulfate were added. The mixture was heated at 100 °C for 70 min. After heating, the sample was washed with 2 L of distilled water for 12 min at 70 °C and immersed in acetone for 10 min. The samples were dried for 10 min and then further dried in a dry oven for 48 h at 60 °C before weighing. The levels of NDF and ADF digestibility were calculated based on the following equations [15,18]:NDF, ADF (%) = [dried residue − (filter bag weight × C)]/sample weight × 100.(2)
Correction (C) = dried blank bag weight/original blank bag weight.(3)
NDF, ADF (g) = sample weight (g) × DM (%) × NDF or ADF (%)/10,000.(4)
NDF, ADF digestibility (%) = NDF or ADF (g) of zero time/NDF or ADF (g) of each time.(5)

The nitrogen content (N%) was analyzed using an analyzer (Kjeltec^TM^ 8400, Foss, Denmark) with sulfuric acid and a catalyst. The crude protein (CP) content was calculated by multiplying N% by 6.25.
CP digestibility (%) = 100 × [CP (g) of each time/CP (g) of zero time] × 100.(6)

### 2.2. In Vivo Experiment

#### 2.2.1. Animals and Diets

Eight Korean native steers (average BW = 597.1 ± 50.26 kg; average age = 23.8 ± 0.12 months) were randomly and evenly assigned to two groups, control (without MO) and MO (3% MO supplementation on a dry matter basis), for 186 days. The animals were placed four steers per pen and fed concentrate and straw. The chemical and fatty acid compositions of the experimental diets are shown in Table 1. Water was provided ad libitum. Feed was provided once a day at 0.900 h. As mentioned previously, 3% MO was added to the diets by top dressing. The body weight of each steer was measured every 3 months prior to feeding using a digital weighing indicator (CAS Co. Ltd., Seoul, Korea). The feed intake (FI) was calculated after measuring the amount of feed remaining from the day before. Then, the following equation was used to calculate from pen data to individual data: FI/BW% = 1.2425 + 1.9218 × net energy for maintenance (NEm)−0.7259 × (NEm)^2^ [19].

#### 2.2.2. Blood Collection and Analysis

Blood samples were taken blood cell counting [20], and metabolite analysis [21] was performed from each cow via the jugular vein before feeding on the first day at 0, 90, and 180 days. In brief, blood was collected into EDTA-containing tubes (Becton Dickinson, Franklin Lakes, NJ, USA) and then subjected to a complete blood cell count test using an HM2 (VetScan HM2 Hematology System, Abaxis, Union City, CA, USA). For the serum samples, blood was collected in heparin-containing tubes (Becton Dickinson) and centrifuged at 3000 rpm for 15 min to extract the serum. The samples were then used to determine the blood metabolite concentrations using a chemical analyzer (Furuno CA-270, Nishinomiya, Japan).

#### 2.2.3. Carcass Traits and Sample Preparation

After slaughtering, we further measured the effect of 3% MO supplementation on the carcass properties using the loin (longissimus thoracis) samples. After approximately 12 h of fasting, the steers were slaughtered following the commercial slaughtering procedure at a commercial slaughterhouse (Cheongju, Korea). Following 24 h of chilling, the carcass weight, yield grade (carcass weight, back fat thickness, and ribeye area) and quality grade (marbling score, meat color, fat color, firmness, and maturity) were evaluated by an official quality grader via the Korean Carcass Grading System of Korea Institute of Animal Products Quality Evaluation (KAPE, 2019). The quality grade evaluation was carried out on the surfaces of longissimus thoracis muscles in the 13th rib section.

#### 2.2.4. Meat Quality

Each longissimus thoracis sample was cut and vacuum-packed into a polyethylene bag at 24 h postmortem and subsequently transported to the laboratory. Samples were stored in a chilled room at 4 °C overnight. The longissimus thoracis samples were sliced to observe the color, pH, cooking loss, shear force, and fatty acid content. The samples were allowed to bloom for 30 min.

Color was determined using a colorimeter (CR-210, Minolta, Tokyo, Japan; illuminate C, calibrated with white plate, L* = +97.83, a* = –0.43, b* = +1.98). The L* (lightness), a* (redness), and b* (yellowness) values were measured on the surface of meat. All cthe onditions were assessed in triplicate.

The pH of each meat sample was measured using a portable pH meter (HI98163, Hanna Instrument, Woonsocket, RI, USA), which was calibrated in buffers (pH 4.0 and 7.0) at room temperature. The pH probe was 20 mm deep into the samples. Measurements were made at three locations in each meat sample.

The cooking loss of each sample was measured using a previously described procedure [22]. Each sample was initially cut and weighed and then packed in closed polyethylene bags and immersed in a temperature-controlled water bath (80 °C) until the core temperature reached 75 °C. Cooked samples were transferred to a cooled water bath until reaching equilibrium and then reweighed. The cooking loss was calculated as the difference in weight before and after cooking as follows:Cooking loss (/100 g) = [weight of raw sample (g) − weight of cooked sample (g)]/weight of raw sample (g) * 100.(7)

The sample preparation to determine the shear force was similar to the cooking loss procedure. Meat samples 1 cm in diameter were cut and tested at room temperature with a texture analyzer (TA-XT2i, Stable Micro Systems, Surrey, UK). Each sample of meat was cut perpendicular to the direction of the muscle fibers. The conditions of the texture analysis were as follows: pre-test speed 2.0 mm/s, post-test speed 8.0 mm/s, load cell 25 kg, maximum load 2 kg, head speed 2.0 mm/s, distance 20.0 mm, and force 5 g. The maximum force at the calculated peak was determined.

Digital color images of beef obtained from 8 steers were used for the image analysis of marbling characteristics. A mirror-type digital camera (D3200, Nikon Co., Tokyo, Japan) was used to photograph the marbling of each sample surface (at the 13th thoracic vertebrae). A strobe (Nikon Co.) was used to prevent irregular reflection on the surface of each sample. An image analysis was performed using the Beef analyzer II (Hayasaka Ricoh Co. Ltd., Sapporo, Hokkaido, Japan), following a previously reported method [23]. Digital color images were converted to binarized images to analyze the marbling characteristics. After binarization, marbling flecks were classified into two forms: big (>0.5 cm^2^) and small (0.01 to 0.5 cm^2^) flecks. However, the smallest particles (<0.01 cm^2^) were not used for the image analysis of marbling characteristics. After calculating the number of marbling flecks and the marbling area, these factors were used to measure the marbling area ratio, coarseness (C), fineness (F), and F/C ratio. The marbling area ratio, the C index, and the F index were calculated following a previously reported method [24].

#### 2.2.5. Fatty Acid Profiles of Meat

The lipid content of each sample was determined using a solvent mixture of chloroform:methanol (2:1, *v/v*) as described in a previous study [25]. FAME Mix standard (Sigma, St. Louis, MO, USA) was the standard used to determine the fatty acid content and fatty acid ratio. The measurement equipment used was a gas chromatograph/FID (7890B GC System, Agilent Technologies), and the analysis conditions followed those used in a previous study [26]: a Sp-2560 capillary column (dimensions: 100 m × 0.25 mm × 0.2 μm, film thickness); injection: split 30:1, heater 255 °C, pressure 32.64, total flow 39.5 mL/min, split flow 36.0 mL/min, injection volume 1.0 μL, helium 1.2 mL/min as the carrier gas; oven program: 70 to 100 °C for 5/min (Hold 2 min), 100 °C to 175 °C for 10/min (hold 40 min), and then 175 to 225 °C for 5/min (hold 40 min); detector: FID System, heater 260 °C (H_2_ flow: 40 mL/min, air flow: 400 mL/min). Conjugated linoleic acid, omega-3 (C18:3n3, C20:3n3, C20:5n3, and C22:6n3), and omega-6 (C18:2n6t, C18:2n6c, C18:3n6, C20:3n6, and C20:4n6) fatty acids were used as standards (Sigma).

### 2.3. Statistical Analysis

The results of the in vitro experiment were analyzed as a one-way ANOVA using the JMP pro 14 (SAS Institute, Cary, NC, USA). In the in vivo experiment, the JMP 7.0 (SAS Institute) program was used for all statistical analyses. Comparisons between two groups were evaluated using Student’s *t*-test. The significance level was declared at *p* < 0.05 and a tendency was declared at 0.05 ≤ *p* < 0.1.

## 3. Results

### 3.1. In Vitro Experiment

We investigated the pH, ammonia-N and VFA contents and digestibility of ruminal fluid by adding various levels of MO (Table 2). Supplementation with 1% to 4% MO did not affect the ruminal pH. Moreover, there were no differences in the ammonia-N and VFA contents among the treatment groups. 

The digestibility of DM and CP were not different between the control group and all treatment groups (Table 3).

Compared to the control, there was no difference in the digestibility of NDF and ADF with 1% to 3% MO, but the group supplemented with 4% MO showed decreased digestibility of NDF and ADF (*p* < 0.05). Considering these results, we determined 3% MO as the optimal dosage for the in vivo experiment.

### 3.2. In Vivo Experiment

#### 3.2.1. Growth Performance

The growth performance is shown in Table 4. The average daily gain (ADG) was increased from days 1 to 90 (*p* = 0.007) in the 3% MO group. 

MO supplementation also showed a tendency to increase the ADG throughout the entire experiment (*p* < 0.1). There were no differences in the BW or FI between the two groups throughout the entire experiment period (total average; day 1 to 180; Table 4). As a consequence, the feed efficiency (FE) increased from days 1 to 90 and for the entire experiment period (*p* < 0.05) in the 3% MO group.

#### 3.2.2. Blood Parameter Analysis

Blood collected from veins of steers was analyzed to determine the hematological (Appendix A) and biochemical parameters (Table 5). 

For the hematological parameters, there were no significant differences between the two groups throughout the entire experiment (Appendix A). The biochemical analysis revealed increases in the levels of glucose (GLU; *p* = 0.041) and high-density lipoprotein-cholesterol (HDL-C; *p* = 0.033) in the 3% MO group on day 90 compared to the control. In addition, the total cholesterol (TCHO) and non-esterified fatty acid (NEFA) contents on day 90 tended to be higher (*p* < 0.1) in the 3% MO group compared with the control group. However, there was no difference in the triglyceride (TG) content between the two groups throughout the entire experiment period.

#### 3.2.3. Carcass Traits

Carcass traits were evaluated over 180 days (Table 6). There were no differences in meat yield or quality traits between the two groups.

#### 3.2.4. Meat Quality

The meat quality results are shown in Table 6. Compared with the control, the 3% MO group showed a tendency for the pH of the meat to increase (*p* < 0.1). The meat from the 3% MO group had a higher (*p* < 0.05) lightness (L*) and tended to be higher (*p* < 0.1) in yellowness (b *) compared with the control. 

There was no difference in redness (a *). On the other hand, the shear force was lower (*p* < 0.05) for steers fed 3% MO than for the controls. Supplementation with 3% MO did not affect the cooking loss or marbling characteristics.

#### 3.2.5. Meat Fatty Acid Profile

The fatty acid composition in the subcutaneous and intramuscular fat of the two groups was investigated (Table 7 and Table 8). Compared to the control group, the concentrations of C15:0, C18:1n9t, and C20:1 fatty acid increased (*p* < 0.05) in the subcutaneous fat of the 3% MO treated group. Moreover, the omega-3 fatty acid (C22:6n3) content increased (*p* < 0.001), and the omega-6 fatty acid (C20:4n6) content decreased (*p* = 0.013). The 3% MO group showed a tendency for C20:2 to decrease (*p* < 0.1).

As shown in Table 8, the concentrations of C17:1 and C18:3n6 fatty acids and omega-3 fatty acids (C18:3n3 and C20:3n3) increased (*p* < 0.05), and the concentrations of C15:0, C16:1, C18:1n9t, and C20:5n3 fatty acids tended to increase (*p* < 0.1) in the intramuscular fat of the 3% MO group. 

Also, a decrease in C22:0 fatty acid was observed (*p* < 0.05). As a result, the level of omega-3 fatty acid increased (*p* < 0.5), and the omega-6 / omega-3 ratio tended to decrease (*p* < 0.1).

## 4. Discussion

### 4.1. In Vitro Experiment

Ruminal fluid pH and the ammonia-N content were not affected in any of the treatment groups (Table 2). Benchaar et al. [7] reported similar results when they added 2%, 3%, and 4% linseed oil to dairy cows. Benchaar et al. [7] suggested that the level of LO supplementation was not high enough to have an effect, and for that reason, there were no changes in ruminal metabolism. Fat supplementation has toxic effects on cellulolytic microbials and digestion, which leads to microbial changes [27]. These microbial changes shift the fermentation pattern to propionate, while the concentrations of acetate and butyrate decrease [11]. In this study, the total VFA and acetate contents tended to decrease in 4% MO, but there was no significant difference in the amount of VFA between the control and MO treatment groups (Table 2). Palmquist et al. [28] reported that the rumen-protected fat maintains normal rumen fermentation. In agreement with a previous study, we suggest that there was no difference in the VFA content, as there was no change in the fermentation pattern [28]. On the contrary, the digestibility of NDF and ADF decreased following supplementation with 4% MO (Table 3; *p* < 0.05). According to previous studies [27], fat supplements coat the rumen microbials, resulting in reduced digestion. In addition, microbials attach to the feed particles to digest fiber, but when fat supplements are added, the fat attaches to the feed particles and interferes with the activity of microbial enzymes [27]. Considering these findings, we considered that with 4% MO, the oil from the MO was eluted, negatively affecting ruminal microbials and resulting in reduced NDF and ADF digestibility (*p* < 0.05). Therefore, we suggest that supplementation with 4% MO led to an overdose of MO that negatively affected fiber digestion and that 3% MO is the optimal dosage for rumen fermentation and digestibility.

### 4.2. In Vivo Experiment

Fat supplements increase the energy density of feed and improve the feed efficiency, but the increased energy density causes a decrease in the FI [12]. Moreover, the decreased FI leads to decreases in animal performance parameters, such as the ADG and the FE. Previous studies noted decreases in the FI and the ADG when steers were fed fat supplements [13]. As mentioned earlier, one of the reasons for the decrease in feed intake is the toxic effect caused by fat supplements coating the rumen microbials and feed. However, in the present experiment, ADG and FE increased (Table 4; *p* < 0.05). Doreau and Chilliard [29] noted that the absence of a negative effect on rumen digestion may be due to the presence of hydrogenated Ca salts. Thus, we suggest that because the MO is coated with hydrogenated palm oil, it did not cause decreases in FI, and ADG, whereas the FE was increased by the increased energy density from the MO. This experiment was performed from March (spring) to September (early fall), so there were seasonal differences. The average daily gain only increased from days 1 to 90 (March to June; THI 58) and not days 91 to 180 (June to September; THI 73). Animal performance depends on a variety of factors such as the FI, environmental conditions, and management system. Therefore, the reasons for observing inconsistent results might be the influence of seasons between days 1 to 90 and days 91 to 180. If there is no adverse effect on rumen fermentation, animals can receive energy from fat supplementation, and the increased energy intake has positive effects on performance.

Hematological parameters were measured for the purpose of investigating the stability of experimental supplementation in this study (Appendix A). There were no differences in the white blood cells, red blood cells, hemoglobin, hematocrit, mean corpuscular volume, mean corpuscular hemoglobin, or mean corpuscular hemoglobin concentration between the two groups. These results indicate that feed with 3% MO had no negative effect on the immunological response in steers. The results of the biochemical analysis are shown in Table 5. Arave et al. [30] reported that the GLU content, which changes in response to energy intake, increased in serum when animals were fed high-energy diets. Consistent with this experiment, the GLU content increased (*p* < 0.05) on day 90 in the 3% MO group. In addition, Arave et al. [30] reported that the blood cholesterol concentration reflects the overall nutritional intake condition, similar to the GLU content. As expected, on day 90, the TCHO tended to be higher (*p* < 0.1) in the 3% MO group. The addition of fat is known to increase the blood cholesterol content. In addition, the results of this experiment show that the increased HDL-C content contributed to TCHO. The content of non-esterified fatty acids also tended to increase (*p* < 0.1) on day 90. Dryden and Marchello [31] reported that an increase in dietary fat was associated with an increase in serum NEFA. Relling and Reynolds [32] also suggested that supplementation with fat increased the NEFA due to the increase in fat absorption. Consistent with previous studies [12], the HDL-C increased (*p* < 0.05) in the 3% MO group. However, these changes were only observed on day 90. Whether these are the results of a reaction due to other factors needs further research.

There were no differences in carcass yield traits or quality traits (Table 6). In agreement with this experiment, Suksombat et al. [8] showed that dietary palm oil and linseed oil, regardless of the type and level of supplementation, do not affect the carcass traits. Also, these results are similar to those of Castro et al. [33], who reported no difference in carcass traits following supplementation with palm oil, olive oil, or soybean oil in Blonde D’ Aquitaine steers. However, Song et al. [34] found that supplementation with soybean plus fish oil tended to increase (*p* = 0.08) the meat color score, and supplementation with soybean oil plus monensin tended to increase (*p* = 0.07) the texture score. These inconsistent results might have occurred due to the difference between the types of supplement and species used in this experiment and previous studies.

The pH of meat is the basis for meat quality evaluation, because it affects the quality of meat, as assessed by factors such as the tenderness, water holding capacity, and meat color. The pH values of the control and 3% MO groups were 5.4 and 5.5. Samples from the group supplementation with 3% MO tended to have higher pH values compared with those from the control group (Table 6). The optimal pH of meat is considered to be 5.4 to 5.8 [35]; thus, in this study, meat from the control and MO supplementation groups had normal pH values. Meat color is one of the factors influencing consumer purchase, and the color values for L* (Lightness), a* (redness), and b* (yellowness) are used to measure meat color. The variation ranges are 3 3 to 41, 11.1 to 23.6, and 6.1 to 11.3 (L*, a*, b*, respectively) [36]. In this experiment, the L* of meat decreased (*p* = 0.042), and the b showed a tendency to increase in the 3% MO group (Table 6). There was no difference in a * values between the two groups. Considering the variation range of each value, the values for L*, a*, and b* in the MO group were within the normal ranges. Meat color depends on age, weight, and nutritional status. Fat supplementation is associated with problems of reducing elements of the meat quality such as taste, meat color, and oxidation resistance, because an increase in the degree of unsaturation promotes oxidation. Fat oxidation changes the color of meat due to red oxymyoglobin being converted to brown metmyoglobin, and this reaction is usually accompanied by the production of a foul odor. In this experiment, however, we proved that 3% MO did not negatively affect the meat color, because the meat color factors were within the accepted ranges. Meanwhile, there was no difference in cooking loss between the two groups (Table 6). Consistent with this study, Wistuba et al. [37] reported that there was no effect on cooking loss in Angus crossbred steers fed 3% fish oil. Pukrop et al. [38] also reported no difference in cooking loss in Angus-Simmental crossbred steers fed 1 g/steer essential oils. The tenderness of meat is considered to be related to the shear force. Kook et al. [39] reported that shear force decreased in early finishing Korean cattle bulls and steers fed 5% fish oil (*p* < 0.001). They found that fish oil affected the fatty acid composition of meat, and the change in fatty acid composition altered the physical properties of meat, which affected the shear force [39]. It has been reported that changes in the fatty acid composition through supplementation or feed affect the tenderness of meat. In addition, the melting point of fatty acids affects the hardness of adipose tissue [40]. Therefore, the changes in fatty acids composition following MO supplementation may have decreased the shear force (*p* < 0.05) in the 3% MO group (Table 6). The marbling characteristics are related to the palatability. It has been suggested that the shear force with F of marbling is lower than the C of marbling [41]. Nakahashi et al. [42] investigated the concentration of monounsaturated fatty acids (MUFA) according to the size of marbling particles in meat (small < 0.4 cm^2^, medium 0.4 to 2.0 cm^2^, large > 2.0 cm^2^). They showed that larger marbling particles were associated with a greater MUFA concentration. In this experiment, there was no difference in marbling characteristics or the level of MUFA in meat in the 3% MO group. However, there have not been many studies on the relationship between fat supplementation and marbling characteristics, and therefore, further research is needed.

We further determined the fatty acid compositions of subcutaneous- and intramuscular fat in steers (Table 7 and Table 8). In terms of the fatty acid composition in subcutaneous fat (Table 7), no differences in saturated fatty acids (SFA), MUFA, and polyunsaturated fatty acids (PUFA) or the omega-6/omega-3 ratio were found. However, the most unexpected result of this experiment was an increase (*p* < 0.05) in docosahexaenoic acid (DHA; C22:6n3) in the 3% MO group. This result is supported by a previous finding reported by Kim et al. [43]. They found that the DHA level was increased (*p* < 0.05) with dietary whole flaxseed (WFS), regardless of the level. However, there is no exact explanation for this finding. This might be due to an increase in α-linolenic acid (ALA; C18:3n3) following 3% MO supplementation, and the promotion of an increase in DHA in the subcutaneous fat by ALA. The MO supplement contains a high concentration of ALA, a precursor to DHA. The metabolic pathway of omega-3 and omega-6 fatty acids proceeds step by step by adding a double bond through desaturases and increasing the length of carbon through elongases. Long-chain omega-3 fatty acids are converted from ALA to eicosapentaenoic acid (EPA; C20:5n3) through a process of elongation by elongases and desaturation by desaturases, such as Δ5-desaturase and Δ6-desaturase. Then, eicosapentaenoic acid is converted to DHA through elongation, unsaturation, and β-oxidation process [44]. However, except for these findings, no specific effect of MO on subcutaneous fat was shown. Generally, the effect of supplementation on subcutaneous fat is comparatively small in steers compared with bulls and cows [43].

The composition of fatty acids in the intramuscular fat of steers was investigated (Table 8). The intramuscular fat content of steers fed 3% MO had a higher C18:1n9t concentration than the control group (*p* < 0.05). This result might be related to the presence of stearic acid (C18:0; SA) in the MO supplement. There is a desaturase that acts on SFAs and converts SFAs into MUFAs, and this desaturase converted SA into C18:1n9t [44]. Supplementation with 3% MO increased (*p* < 0.05) the contents of C18:3n6 and ALA in the intramuscular fat. Similar results have been reported by Kim et al. [43] in steers fed 10% or 15% WFS. Additionally, many previous studies [45] have reported an increase in ALA when linseed was added. Linoleic acid (LA; 18: 2n6) and ALA are essential fatty acids that cannot be synthesized and must be consumed as feed. When fat supplementation was conducted, as mentioned earlier, many shifts in the fatty acid composition occurred through enzyme systems, such as desaturation and elongation [44]. As a result of desaturation and elongation, the fed LA was unsaturated with ∆6-desaturase, resulting in an increase in C18:3n6. In addition, ALA increased the EPA content through ∆6-desaturase, elongase, and ∆5-desaturase in intramuscular fat. Supplements containing ALA always elevated EPA [46]. However, DHA, which existed in the subcutaneous fat, was not observed in the intramuscular fat (Table 8). In general, there is a metabolic difference between subcutaneous fat and intramuscular fat in ruminants. Smith and Crouse [47] suggested that the main precursor of subcutaneous fat is acetate and the main precursor of intramuscular fat is likely to be glucose. Although ALA is a precursor of long-chain omega-3 fatty acids, there is limited conversion. Noci et al. [48] reported that conversion is affected for a variety of factors. In short, it is due to the activity of low desaturase and competition with other long-chain fatty acids. When ALA is converted to DHA, it is controlled by a complex enzyme system that also acts on other long-chain fatty acids. Thus, due to the low conversion efficiency, the conversion from ALA to DHA was not observed consistently. Considering all of these results, the level of omega-3 fatty acid was elevated (*p* < 0.05) and the omega-6/omega-3 ratio tended to decrease (*p* = 0.05) in the 3% MO group. Overall, the supplementation of steers with 3% MO caused many shifts in fatty acid composition. In general, intramuscular fat is affected by several internal and external factors such as age, gender, species, castration, temperature, and nutrition. Therefore, further studies are needed to clarify the effects of these factors on intramuscular fat in steers.

## 5. Conclusions

Feeding with 3% MO, a rumen protected fatty acid enriched with omega-3 fatty acid, improved the growth performance of Korean native steers and decreased the shear force produced by the muscles. By increasing the level of omega-3 fatty acids in intramuscular fat, MO supplementation tended to decrease the omega-6/omega-3 ratio and improve the meat quality. In conclusion, it was demonstrated that supplementation with 3% MO increased the productivity of Korean native steers by producing healthful beef enriched with omega-3 fatty acids.

## Figures and Tables

**Table 1 animals-11-01253-t001:** Chemical and fatty acid compositions of the steers’ diets.

Items	Straw	Concentrate	MO ^1^
	Chemical composition, %		
Dry matter	12.69	11.83	0.01
Crude protein	5.17	16.62	0.12
Ether extract	1.84	3.92	94.28
Crude fiber	36.30	5.36	-
Crude ash	11.07	7.11	4.98
Neutral detergent fiber	41.37	24.68	-
Acid detergent fiber	69.76	7.49	-
Ca	0.34	0.78	
P	0.1	0.44	
	Fatty acids, %		
8:0	0.64	0.42	
10:0	1.11	0.40	
12:0	1.08	5.83	
14:0	35.24	2.01	
15:0	2.06		
16:0	1.06	14.46	42.66
16:1	7.71	0.14	
17:0	23.83	0.08	
18:0	12.87	2.56	14.14
18:1n9c	3.16	24.88	17.70
18:2n6c	2.78	43.75	5.92
18:3n3	0.48	4.24	18.31
20:0	2.58	0.41	
20:1	0.87	0.32	
22:0	1.53	0.25	
22:1n9	2.97	0.04	
22:2			
24:0		0.22	

^1^ MO = rumen-protected microencapsulated supplement comprising linseed oil, vitamin E, rosemary extract, and hydrogenated palm oil.

**Table 2 animals-11-01253-t002:** Effect of 3% rumen-protected microencapsulated supplement (MO) on pH and ammonia-N and volatile fatty acid (VFA) contents in the in vitro rumina fluid.

Item	CON ^1^	MO Concentration (%) ^2^				SEM ^3^	*p*-Value
		1	2	3	4		
	0 h						
pH	6.80	6.83	6.83	6.82	6.85	0.010	0.652
NH_3_-N	36.61	38.75	38.73	40.65	39.41	0.751	0.694
	48 h						
pH	6.61	6.66	6.41	6.64	6.64	0.023	0.980
NH_3_-N	156.20	146.68	149.08	148.01	136.60	2.840	0.318
	48 h						
Total VFA (mM)	100.78	106.10	109.64	107.91	91.31	2.445	0.089
Acetate	65.48	70.40	72.85	70.69	59.12	1.772	0.068
Propionate	19.52	20.10	20.78	20.85	17.19	0.515	0.124
iso-butyrate	1.07	1.07	1.09	1.08	1.01	0.015	0.510
Butyrate	12.24	12.13	12.41	12.76	11.64	0.209	0.598
iso-valerate	1.10	1.08	1.09	1.09	1.06	0.012	0.852
Valerate	1.37	1.33	1.40	1.43	1.30	0.018	0.145
A:P ratio	3.36	3.50	3.50	3.40	3.44	0.027	0.379

^1^ CON, no MO supplementation to basal diet. ^2^ MO, 1% to 4% MO supplementation to basal diet. ^3^ SEM, standard error of the mean.

**Table 3 animals-11-01253-t003:** Effect of 3% rumen-protected microencapsulated supplement (MO) on digestibility in the in vitro ruminal fluid.

Digestibility, %	CON ^1^	MO Concentration (%) ^2^				SEM ^3^	*p*-Value
		1	2	3	4		
	48 h						
Dry matter	55.57	55.40	57.29	56.19	57.04	0.484	0.714
NDF	23.15 ^b^	23.71 ^a,b^	24.68 ^a^	23.58 ^a,b^	21.78 ^c^	0.273	<0.001
ADF	21.37 ^a,b^	21.35 ^a,b^	22.17 ^a^	21.69 ^a,b^	20.06 ^b^	0.242	0.040
Crude protein	43.17	44.20	47.71	44.18	46.27	0.619	0.104

^1^ CON, no MO supplementation to basal diet. ^2^ MO, 1% to 4% MO supplementation to basal diet. ^3^ SEM, standard error of the mean. ^a–c^ Different superscripts represent difference among treatments (*p* < 0.05).

**Table 4 animals-11-01253-t004:** Effect of 3% rumen-protected microencapsulated supplement (MO) on growth performance in steers.

Items	CON	3% MO ^1^	*p*-Value
	Body weight, kg/head		
Day 0	596.75	597.38	0.988
Day 90	645.38	667.88	0.565
Day 180	689.62	721.62	0.540
	Feed intake, kg/head		
Days 1 to 90	9.08	9.19	0.867
Days 91 to 180	8.40	8.65	0.621
Total Average	8.74	8.93	0.768
	Average daily gain, kg/d		
Days 1 to 90	0.53	0.77	0.008
Days 91 to 180	0.48	0.58	0.593
Total Average	0.50	0.68	0.071
	Feed efficiency ^2^		
Days 1 to 90	5.84	8.54	0.030
Days 91 to 180	5.70	6.63	0.642
Total Average	5.75	7.57	0.037

Values are express as means (*n* = 4). ^1^ 3% MO, 3% MO supplementation to concentrate. ^2^ feed efficiency = average daily gain (ADG)/daily feed intake × 100.

**Table 5 animals-11-01253-t005:** Biochemical analyses of blood in steers supplemented with concentrate containing 3% rumen-protected microencapsulated supplement (MO).

Items	CON	3% MO ^1^	*p*-Value
	Glucose, mg/dL		
Day 0	69.75	73.5	0.224
Day 90	34.75	47.5	0.041
Day 180	54.00	63.25	0.463
	Triglycerides, mg/dL		
Day 0	15.00	13.75	0.502
Day 90	16.75	15.00	0.768
Day 180	23.75	24.25	0.933
	Total cholesterol, mg/dL		
Day 0	163.50	205.50	0.121
Day 90	108.25	147.75	0.073
Day 180	148.75	158.75	0.586
	High-density lipoprotein cholesterol, mg/dL		
Day 0	87.00	120.50	0.167
Day 90	50.75	74.50	0.033
Day 180	78.25	87.75	0.543
	Non-esterified fatty acid, mg/dL		
Day 0	170.5	222.25	0.484
Day 90	137.5	191.5	0.056
Day 180	195.0	202.0	0.862

Values are express as means (*n* = 4). ^1^ 3% MO, 3% MO supplementation to concentrate.

**Table 6 animals-11-01253-t006:** Effect of 3% rumen-protected microencapsulated supplement (MO) on the carcass traits, meat quality, and marbling characteristics of steers.

Items	CON	3% MO ^1^	*p*-Value
	Carcass yield traits		
Back fat thickness, mm	14.0	17.5	0.309
Eye muscle area, cm^2^	90.0	87.5	0.532
Carcass weight, kg	419.3	441.5	0.524
Meat quantity index	64.3	61.3	0.298
	Quality traits ^2^		
Marbling score	4.5	5.3	0.477
Meat color	4.8	5.0	0.356
Fat color	3.0	3.0	-
Texture	1.0	1.3	0.356
Maturity	2.0	2.0	-
Quality grade	3.0	3.5	0.356
	Meat quality index		
pH	5.45	5.51	0.054
Meat color		147.75	0.073
L *	42.00	39.79	0.042
a *	18.39	19.84	0.182
b *	6.41	8.44	0.062
Cooking loss, %	29.43	28.27	0.122
Shear force, kg	21.79	17.78	0.004
	Marbling characteristics		
Marbling percent, %	24.72	22.26	0.304
Fineness (F)	2.44	2.32	0.640
Coarseness (C)	0.19	0.18	0.685
F/C	12.78	13.11	0.795

Values are express as means (*n* = 4). ^1^ 3% MO, 3% MO supplementation to concentrate. ^2^ Evaluated QG based on the grading standard: Marbling score standard: 1 = devoid, 9 = very abundant; meat color standard: 1 = bright red, 7 = dark red; fat color standard: 1 = white, 7 = yellowish; texture: 1 = firm, 3 = soft; maturity: 1 = young, 3 = mature; quality grade: 1+ grade (good) = 4, 1 grade = 3, 2 grade = 2, 3 grade (bad) = 1.

**Table 7 animals-11-01253-t007:** Effect of a 3% rumen-protected microencapsulated supplement (MO) on the fatty acid composition (%) of subcutaneous fat in steers.

Items	CON	3% MO ^1^	*p*-Value
C10:0	0.05	0.05	0.861
C12:0	0.11	0.11	0.963
C14:0	3.82	4.12	0.301
C14:1	2.52	2.53	0.974
C15:0	0.39	0.44	0.043
C16:0	24.10	25.14	0.278
C16:1	8.86	9.18	0.657
C17:0	0.45	0.49	0.363
C17:1	1.03	1.03	0.981
C18:0	5.92	5.69	0.728
C18:1n9t	0.34	0.48	0.032
C18:1n9c	47.67	45.76	0.177
trans-11 C18:1	0.83	0.78	0.725
C18:2n6	0.44	0.42	0.293
C18:2n6c	2.34	2.70	0.134
C20:0	0.02	0.02	0.237
C18:3n6	0.05	0.05	0.460
C20:1	0.17	0.25	0.024
C18:3n3	0.48	0.38	0.142
C20:2	0.06	0.05	0.082
C22:0	0.13	0.12	0.451
C22:1n9	0.01	0.02	0.480
C20:3n3	0.13	0.12	0.622
C20:4n6	0.02	0.01	0.013
C20:5n3	0.06	0.06	0.592
C22:6n3	0.00	0.01	<0.001
w3, %	0.67	0.57	0.246
w6, %	2.84	3.18	0.173
w6:w3	4.48	5.56	0.253
SFA ^2^, %	35.00	36.18	0.320
MUFA ^3^, %	61.43	60.02	0.262
PUFA ^4^, %	3.58	3.80	0.241
SFA/UFA ^5^	0.54	0.57	0.321

Values are expressed as means (n = 4). ^1^ 3% MO, 3% MO supplementation to concentrate. ^2^ SFA = Saturated fatty acids, ^3^ MUFA = Mono-unsaturated fatty acids, ^4^ PUFA = Poly-unsaturated fatty acids. ^5^ UFA = Unsaturated fatty acids.

**Table 8 animals-11-01253-t008:** Effect of 3% rumen-protected microencapsulated supplement (MO) on the fatty acid composition (%) of intramuscular fat in steers.

Items	CON	3% MO ^1^	*p*-Value
C10:0	0.06	0.06	0.470
C12:0	0.08	0.08	0.872
C14:0	3.29	3.61	0.388
C14:1	0.79	1.01	0.313
C15:0	0.29	0.35	0.074
C16:0	27.72	28.00	0.733
C16:1	3.58	4.61	0.061
C17:0	0.70	0.73	0.528
C17:1	0.60	0.78	0.033
C18:0	13.31	10.93	0.125
C18:1n9t	0.18	0.27	0.051
C18:1n9c	44.5	44.15	0.842
trans-11 C18:1	0.34	0.40	0.531
C18:2n6t	0.22	0.25	0.232
C18:2n6c	3.01	3.33	0.322
C20:0	0.03	0.02	0.061
C18:3n6	0.16	0.21	0.042
C18:3n3	0.47	0.61	0.023
C20:2	0.017	0.01	0.746
C22:0	0.18	0.15	0.024
C20:3n6	0.39	0.34	0.332
C22:1n9	0.01	0.01	0.834
C20:3n3	0.01	0.02	0.022
C20:4n6	0.02	0.02	0.152
C20:5n3	0.048	0.054	0.047
w3, %	0.53	0.68	0.022
w6, %	3.80	4.15	0.334
w6:w3	7.18	6.14	0.053
SFA ^2^, %	45.65	43.93	0.341
MUFA ^3^, %	50.01	51.23	0.540
PUFA ^4^, %	4.34	4.84	0.218
SFA/UFA ^5^	0.84	0.78	0.343

Values are express as means (*n* = 4). ^1^ 3% MO, 3% MO supplementation to concentrate. ^2^ SFA = saturated fatty acid, ^3^ MUFA = mono-unsaturated fatty acid, ^4^ PUFA = poly-unsaturated fatty acid. ^5^ UFA = unsaturated fatty acid.

## Data Availability

Not applicable.

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
