# Peer review of "Effect of a Rumen-Protected Microencapsulated Supplement from Linseed Oil on the Growth Performance, Meat Quality, and Fatty Acid Composition in Korean Native Steers"

_animals, 2021, doi:10.3390/ani11051253_

Round 1

Reviewer 1 Report

Effect of rumen-protected microencapsulated supplement from linseed oil on growth performance, meat quality, and fatty acid composition in Korean native steers

The paper addresses the very current field of changing animal products into healthy humane food by means of proper livestock nutrition. The manuscript has it’s pros and cons.

The advantages:

The paper is properly written, well-structured, and supported with good references. The methodology is suitable and the description of used methods is very good. From a substantive point of view (scientific, methodological), everything is very good.

The disadvantages:

The parts that require revision are: the goal of the study and conclusions.

Introduction – goal of the study

What has me worried while reading the manuscript is that the Authors were unable to point why did they conduct this study (though lots of research were done in this field) . Claiming that :”Although many previous studies have investigated rumen-protected supplementation, the effects on performance, meat quality, and fatty acid composition in steers require further research”. Why further research are required? Why were you looking for when you decided for this experiment? What for did you repeat a study that was already conducted by other authors if you did not have any special expectations?

Conclusions

  • ‘ By increasing the level of omega-3 fatty acids in intra-506 muscular fat, MO supplementation tended to decrease the omega-6/omega-3 ratio and improve the meat quality.’ – how did the meat quality improved, be more precise. After studying the meat quality measures, a really significant change can be found only for the shear force, all the other shifts in quality are very slight (pH, color). If only the shear force clearly changed, do you thing that MO supplementation may be important in high quality beef production?
  • Did you find anything in your results, that you did not expect after reading the available literature?

Reviewer 2 Report

Dear Authors, I have revised the abovementioned manuscript. Design new feeding strategies to improve growth performance and meat fatty acids content is a very important topic to improve the feasibility of the beef production system and, at the same time, to increase consumers' interest regarding the meat industry. Therefore, the authors' contribution to literature is appreciated. The manuscript is well structured and detailed. However, in my opinion, some parts need to be revisited. My major criticism regards the in vitro study. Based on the purposes stated by the authors (see L 61-63), the in vitro study should serve as a preliminary test aimed at establishing the optimal dose of microcapsules. Therefore, from my point of view, I would have expected to see more doses tested (before you get to read the introduction). Well, as I can read, it is not possible to deduce this information from the materials and methods. Additionally, the simple summary, which I consider extremely succinct and scarcely explanatory. This part of the manuscript, in fact, should also be addressed to non-specialized users, who can benefit from a greater explanation of the assumptions on which the research is based and, on the health-related effects of the results. Apart from the above, further suggestions are detailed below. Hoping to have contributed to improving the manuscript quality, I wish the authors a good job.

Specific comments

L 33 (keywords): in my opinion, “meat quality” and “growth performance” could be inverted, thanks.

L 50: forage feeding effects is well addressed by https://doi.org/10.3168/jds.2018-14710 and https://doi.org/10.1016/j.jclepro.2017.03.078, that I warm recommended to list as references.

L 59: please, delete “previous”.

L 61: I believe the authors meant in vitro.

L 75-99: authors are invited to consider as stated in general remarks.

L 148: please, replace “food” with “feed”.

L 152-154: the sentence seems unclear. Authors are invited to better specify.

Round 2

Reviewer 2 Report

Dear authors,

taking into account the latest changes, I believe that the manuscript deserves to be published in Animals
Congratulations